# "I pick you choose": Joint human-algorithm decision making in multi-armed bandits

## Abstract

Online learning in multi-armed bandits has been a rich area of research for decades, resulting in numerous "no-regret" algorithms that efficiently learn the arm with highest expected reward. However, in many settings the final decision of which arm to pull isn't under the control of the algorithm itself. For example, a driving app typically suggests a subset of routes (arms) to the driver, who ultimately makes the final choice about which to select. Typically, the human also wishes to learn the optimal arm based on historical reward information, but decides which arm to pull based on a potentially different objective function, such as being more or less myopic about exploiting near-term rewards. In this paper, we show when this joint human-algorithm system can achieve good performance. Specifically, we explore multiple possible frameworks for human objectives and give theoretical regret bounds for regret. Finally, we include experimental results exploring how regret varies with the human decision-maker's objective, as well as the number of arms presented.

## 1 Introduction

Consider the following motivating example:

> Alice has recently moved to a new town and does not know the area yet. She uses a navigation app while driving, which narrows down the thousands of potential routes to a few options for her to choose from. She and the app only get to see the actual driving time of the final route she picks. Because of varying traffic and weather delays, the actual driving times of each route is unpredictable. Both the navigation app and Alice wish to minimize her average travel time. However, they might have different short-term objectives. For example, Alice might be *myopic* and prefer choosing a route that has performed well in the past, rather than exploring a new one. Alternatively, Alice may be adventurous and actively seek out new routes in the hope that there might be quicker than ones she has previously explored. The navigation app uses a generic algorithm that doesn't know Alice's specific objective function. Under what situations can Alice and her navigation app achieve their goal of quickly finding the quickest route?

If Alice's navigation app were able to tell Alice exactly which route she must take, then this problem would reduce to that of *multi-armed bandits* (MAB), a celebrated online-learning paradigm. However, in the driving directions setting, it is unrealistic to assume that the algorithm can force Alice to take a particular route. In human-algorithm collaboration more generally, often the algorithm can provide assistance, but the human makes the final decision. This is the case in other settings as well: a diner trying to find the best restaurant, a doctor trying to find the best treatment, or a teacher trying to find the best pedagogical method. This framework requires a shift in thinking: rather than focus on optimizing the performance of the algorithm alone, the goal is to build an algorithm that maximizes the performance of the human-algorithm system.

For multi-armed bandits, the standard objective is to minimize expected regret, the amount of reward that is missed by not selecting the optimal arm. In human-algorithm multi-armed bandits, some aspects (such as the behavior of the human), are entirely out of our control, and so the system cannot

be completely optimized. Instead, the goal of this paper is *descriptive*: to characterize settings where sublinear regret is possible - and settings where linear regret is unavoidable.

In Section 2, we discuss how our setting and results relate to previous literature in MAB and in human-algorithm collaboration. In Section 3, we formalize the model that we analyze, including multiple different models of human behavior. Section 4 contains theoretical results, such as bounds on expected regret. Specifically, we show that, so long as the human isn't completely myopic (has some weak preference for exploring arms that haven't been frequently pulled), then sublinear regret is achievable. If the human is myopic, then it is unavoidable that the regret includes a linear dependence on time. Section 5 enriches these theoretical results with experimental simulations. These results show that if the human is more myopic than the algorithm, overall regret *decreases* the more arms are shown to the human. On the other hand, if the human is *less* myopic, the opposite is true, and regret *increases* the more arms are shown to the human. Finally, in Section 6 we briefly discuss implications of our work and potential future directions.

## 2    RELATED WORK

### MULTI-ARMED BANDITS

The area of multi-armed bandits is wide enough to admit multiple textbooks Slivkins et al. (2019); Lattimore & Szepesvári (2020). In this section, we will highlight some of the most related papers.

Yue et al. (2012) proposed "dueling bandits", where multiple arms are presented simultaneously and the feedback is noisy binary signal as to which has higher reward. Since this, there has been numerous extensions Sui et al. (2018; 2017b); Komiyama et al. (2015), such as those that allow more than 2 arms to be presented Saha & Gopalan (2018); Agarwal et al. (2020); Sui et al. (2017a). Shivaswamy & Joachims (2015) studies a related problem where the task of the algorithm is to rank a set of items. The human then improves the ranking according to their true utility function, but with some bounded degree of improvement reflecting limits on human rationality. One major difference between dueling bandits and our framework is that we assume feedback is given by a human who is learning about the rewards of the arms themselves, whereas dueling bandits typically assume that responses between the arms are fixed. Additionally, dueling bandits typically involves boolean feedback, where we allow real-valued access to the rewards.

There has also been a series of work looking more specifically at human-algorithm collaboration in bandit settings. Gao et al. (2021) learns from batched historical human data to develop an algorithm that assigns each task at test time to either itself or a human. Chan et al. (2019) studies a setting similar to ours in that the human is simultaneously learning which option is best for them. However, their framework allows the algorithm to overrule the human, which makes sense in many settings, but not all, such as our motivating example of driving directions. Bordt & Von Luxburg (2022) formalizes the problem as a two-player setting where both the human and algorithm take actions that affect the reward both experience.

Additionally, some work has used the framework of the human as the final decision-maker and studied how to disclose information so as to incentivize them to take the "right" action. Immorlica et al. (2018) studies how to match the best regret in a setting where myopic humans pull the final arm. Hu et al. (2022) studies a related problem with combinatorial bandits, where the goal is to select a subset of the total arms to pull. Bastani et al. (2022) investigates a more applied setting where each human is a potential customer who will become disengaged and leave if they are suggested products (arms) that are a sufficiently poor fit. Kannan et al. (2017) looks at a similar model of sellers considering sequential clients, specifically investigating questions of fairness. In general, these works differ from ours in that they assume a new human arrives at each time step, and so the algorithm is able to selectively disclose information to them. In our setting, the human may be the same between time steps, and we typically assume that they have access to the same information as the algorithm.

### HUMAN-ALGORITHM COLLABORATION

Studying human-algorithm collaboration is a rapidly growing, highly interdisciplinary area of research. In general, most work focuses on offline learning settings, which differs from our MAB

analysis. Some veins of research are more ethnographic, studying how people use algorithmic input in their decision-making Lebovitz et al. (2021; 2020); Beede et al. (2020); Yang et al. (2018); Okolo et al. (2021). Other avenues work on developing ML tools designed to work with humans, such as in medical settings Raghu et al. (2018) or child welfare phone screenings Chouldechova et al. (2018). Finally, and most closely related to this paper, some works develop theoretical models to analyze human-algorithm systems, such as Rastogi et al. (2022); Cowgill & Stevenson (2020); Bansal et al. (2021a); Steyvers et al. (2022); Madras et al. (2018). Bansal et al. (2021b) proposes the notion of *complementarity*, which is achieved when a human-algorithm system together has performance that is strictly better than either the human or the algorithm could achieve along. Straitouri et al. (2022) studies "conformal prediction" where the algorithm narrows down the the list of possible item labels to a subset, from which the human picks. This formulation is structurally similar to ours, but considers the offline labeling task, rather than online MAB.

## 3 MODEL AND ASSUMPTIONS

### 3.1 MODEL

We assume that there are $N$ arms, each of rewards drawn i.i.d from the distribution $X_i \sim \mathcal{D}_i$, for $i \in [N]$. WLOG, we will order the arms in descending order of expected reward. This means arm 0 has the highest expected reward, and we will set $\Delta_i$ to be the difference between the expected reward of arm $i$ and arm 0. There are two actors, the human ($H$) and the algorithm ($A$). Each of them has access to the same historical information but uses it in different ways. For each time step $t \in [T]$, the algorithm selects a subset of $k \in [1, N]$ arms to present to the human. Among those $k$ presented, the human selects a single final arm $I_t$ to be pulled. Both the human and algorithm observe reward $X_{i,t} \sim \mathcal{D}_i$. Note that for $k = 1$ this reduces to the algorithm selecting the final arm (because the human can only select from those that are presented), while for $k = N$ this reduces to the human making unconstrained selection. Throughout, our goal will be to minimize expected *regret*, or the amount of reward the human-algorithm system misses out on by pulling sub-optimal arms:

$$T \cdot \mu_0 - \sum_{t=0}^{T} \mathbb{E}[\mu_{I_t}] = \sum_{t=0}^{T} \mathbb{E}[\Delta_{I_t}]$$

### 3.2 ALGORITHM AND HUMAN BEHAVIOR

Next, we will describe the assumptions behind how the algorithm and human behave. One standard selection approach we will incorporate is the UCB algorithm, which at time $t$ selects whichever arm maximizes Auer et al. (2002):

$$\hat{\mu}_{i,t} + \alpha_a \cdot \sqrt{\frac{\ln(t)}{n_{i,t}}}$$

for empirical mean $\hat{\mu}_{i,t} = \frac{1}{n_{i,t}} \sum_{s=1}^{t} \mathbb{1}[I_s = i] \cdot X_{i,t}$. Because the UCB algorithm is a standard algorithm for multi-armed bandit settings, we will assume the algorithm $A$ uses some variant of it. However, in this paper, we will explore scenarios where the human $H$ uses multiple different selection rules. For example, we say that $H$ is $(\alpha_h, \delta)$-*myopic* if it selects randomly among the $k$ presented arms with probability $\delta$ and otherwise selects whichever maximizes the UCB algorithm with coefficient $\alpha_h$:

$$\begin{cases} x \sim \text{Unif}[k] & r \sim \text{Unif}[0,1] \leq \delta \\ \text{argmax}_i \left[ \hat{\mu}_{i,t} + \alpha_h \cdot \sqrt{\frac{\ln(t)}{n_{i,t}}} \right] & \text{otherwise} \end{cases}$$

In addition to these objectives, we will assume that the human and algorithm both prefer to pull each arm at least once before pulling any other arm.

## 4 THEORETICAL ANALYSIS

In this section, we will provide theoretical regret bounds for our human-algorithm setting. First, we will show in Lemma 1 that any human with $\delta > 0$ must incur at linear regret (proof in the Appendix). Additionally, this result has a lower bound that is increasing in $k$, the number of arms shown to the human. This suggests that, if the human selects randomly with some nonzero probability, it may be optimal to show as few arms as possible to them. Following this result, we will assume $\delta = 0$ throughout the rest of the paper.

**Lemma 1.** *Any human that selects uniformly at randomly with probability $\delta > 0$ among $k \geq 2$ arms incurrs regret $\Omega(T)$ that is increasing in $k$.*

Next, we will work to bound total regret when $\delta = 0$. We will find it useful to use Lemma 2, which gives a high-probability bound for the relative ordering of the UCB values for two arms.

**Lemma 2.** *Consider arm $i$ with $n_{i,t} \geq \alpha^2 \cdot \frac{\ln(t)}{\epsilon^2 \cdot \Delta_i^2}$, for $\epsilon \in [0, 0.5], \alpha \geq 1$. Then, if any UCB algorithm selecting according to $UCB_{i,t} = \hat{\mu}_{i,t} + \alpha \cdot \sqrt{\frac{\ln(t)}{n_{i,t}}}$. Then, $UCB_{i,t} < UCB_{0,t}$ with probability at least $1 - \frac{4}{t^2}$.*

First, we consider the case where $\alpha_h > 0$, so the human isn't completely myopic in its choice of arm. Theorem 1 gives a regret bound for this scenario. One key area of focus in human-algorithm collaboration in general is how the performance of the joint system compares to that of the human and algorithm separately Bansal et al. (2021b;a). Overall performance could be better than either the human or algorithm (*complementary* performance, as defined inBansal et al. (2021b)), or could be worse, or could be somewhere in between. For Theorem 1, the regret bound involves a $\max(\alpha_h^2, \alpha_a^2)$ term. This effectively means that regret for the human-algorithm system is guaranteed to at least as good as the worse of the two components (human or algorithm).

**Theorem 1.** *Consider the case with $\alpha_h, \alpha_a > 1$. Then, the expected regret is bounded by:*

$$\sqrt{N \cdot T \cdot \ln(T)} \cdot \left(1 + 4 \cdot \max\left(\alpha_h^2, \alpha_a^2\right)\right) + 16 \cdot N$$

*Proof.* First, we will divide the two arms into groups:

1. Group 1 contains arms with $\Delta_i < \sqrt{\frac{N}{T} \cdot \ln(T)}$

2. Group 2 contains arms with $\Delta_i \geq \sqrt{\frac{N}{T} \cdot \ln(T)}$

In order to bound total regret, we can bound regret from group 1 as:

$$\sum_{i \in G_1} n_{i,T} \cdot \Delta_i \leq \sqrt{\frac{N}{T} \cdot \ln(T)} \cdot \sum_{i \in G_1} n_{i,T} \leq \sqrt{\frac{N}{T} \cdot \ln(T)} \cdot T = \sqrt{N \cdot T \cdot \ln(T)}$$

The remaining portion of the proof will bound regret from arms of group 2.

First, we will work to bound the number of times that arm $i$, for $i \neq 0$, can be pulled for all time ($n_{i,T}$).

We will use $I_t$ to indicate the arm pulled at time $t$. Our goal is to bound the expected value of $n_{i,T}$. We will use $n_{i,t}^* = 4 \cdot \max\left(\alpha_h^2, \alpha_a^2\right) \cdot \frac{\ln(t)}{\Delta_i^2}$ to denote a minimum threshold of samples we need from

arm $i$ in order to obtain certain high-probability guarantees.

$$\mathbb{E}[n_{i,T}] = \mathbb{E}\left[\sum_{t=0}^{T} \mathbb{1}(I_{t+1} = i)\right]$$

$$= 1 + \mathbb{E}[\sum_{t=N}^{T} \mathbb{1}(I_{t+1} = i)]$$

$$= 1 + \mathbb{E}\left[\sum_{t=N}^{T} \mathbb{1}\left(I_{t+1} = i | n_{i,t} < n_{i,t}^*\right)\right] + \mathbb{E}\left[\sum_{t=N}^{T} \mathbb{1}\left(I_{t+1} = i | n_{i,t}^*\right)\right]$$

$$\leq n_{i,T}^* + \mathbb{E}\left[\sum_{t=N}^{t} \mathbb{1}\left(I_{t+1} = i | n_{i,t} \geq n_{i,t}^*\right)\right]$$

In the second line, we used the fact that each arm must be pulled at least once. In the third line, we conditioned on the probability of pulling arm $i$ depending on whether it has already been pulled $n_{i,t}^*$ times or not. In the fourth line, we use the fact that arm $i$ can only be pulled $n_{i,T}^*$ times until $n_{i,t} < n_{i,t}^*$ is no longer satisfied.

Next we will bound the second term: the number of times arm $i$ can be selected, given that it has already been pulled at least $n_{i,t}^*$. Here, we will find it useful to rewrite the expectation as the sum of probabilities:

$$\mathbb{E}\left[\sum_{t=N}^{T} \mathbb{1}\left(I_{t+1} = i | n_{i,t} \geq n_{i,t}^*\right)\right] = \sum_{t=N}^{T} P\left(I_{t+1} = i | n_{i,t} \geq n_{i,t}^*\right)$$

Recall that the algorithm uses $UCB_{i,t}^a = \hat{\mu}_i + \alpha_a \cdot \sqrt{\frac{\ln(t)}{n_{i,t}}}$, while the human uses $UCB_{i,t}^h = \hat{\mu}_i + \alpha_h \cdot \sqrt{\frac{\ln(t)}{n_{i,t}}}$. We have set $n_{i,t}^* = 4 \cdot \max\left(\alpha_h^2, \alpha_a^2\right) \cdot \frac{\ln(t)}{\Delta_i^2}$ so that, according to Lemma 2, we have enough samples that with probability at least $1 - \frac{4}{t^2}$: 1) $UCB_{i,t}^a < UCB_{0,t}^a$ and 2) $UCB_{i,t}^h < UCB_{0,t}^h$. Statement 1) means that if arm $i$ is presented to the human, with high probability arm 0 will be as well. Statement 2) means that if both arms are presented to the human, with high probability the human will pick arm 0. If either of these fails, then arm $i$ could get picked.

We can use both of these facts to bound regret from this portion based on the probability that either of these conditions fails:

$$\leq \sum_{t=N}^{T} 2 \cdot \frac{4}{t^2} \leq 8 \cdot 2$$

We can bound the instance-independent regret from this term (for arms in group 2):

$$\sum_{i \in G_2} n_{i,t} \cdot \Delta_i \leq \left(n_{i,T}^* + 16\right) \cdot \Delta_i$$

$$= \left(4 \cdot \max\left(\alpha_h^2, \alpha_a^2\right) \cdot \frac{\ln(T)}{\Delta_i^2} + 16\right) \cdot \Delta_i$$

$$= \sum_{i \in G_2} 4 \cdot \max\left(\alpha_h^2, \alpha_a^2\right) \cdot \frac{\ln(T)}{\Delta_i} + 16 \cdot \Delta_i$$

$$\leq \sum_{i \in G_2} 4 \cdot \max\left(\alpha_h^2, \alpha_a^2\right) \frac{\ln(T) \cdot \sqrt{T}}{\sqrt{N \cdot \ln(T)}} + 16 \qquad \sqrt{\frac{N}{T} \cdot \ln(T)} \cdot \leq \Delta_i \leq 1$$

$$\leq 4 \cdot \max\left(\alpha_h^2, \alpha_a^2\right) \cdot \sqrt{N \cdot T \cdot \ln(T)} + 16 \cdot N$$

Overall, if we add in the regret from arms of group 1, we get a regret bound of:

$$\sqrt{N \cdot T \cdot \ln(T)} + 4 \cdot \max\left(\alpha_h^2, \alpha_a^2\right) \cdot \sqrt{N \cdot T \cdot \ln(T)} + 16 \cdot N$$

$$= \sqrt{N \cdot T \cdot \ln(T)} \cdot \left(1 + 4 \cdot \max\left(\alpha_h^2, \alpha_a^2\right)\right) + 16 \cdot N$$

$\square$

Next, we consider the case where $\alpha_h = 0$, so the human greedily pulls whichever arm has had highest empirical reward (of those presented to it). Theorem 2 gives a regret bound for this scenario. As compared with Theorem 1, note that this bound includes a linear dependence on $T$ that increases with $k$, the number of arms shown to the human. In Lemma 3, we show that linear regret is unavoidable if the human picks purely myopically.

**Theorem 2.** *Consider any human using myopic selection ($\alpha_h = 0$) and $\alpha_a \geq 1$. . Then, the expected regret is bounded by:*

$$\sqrt{N \cdot T \cdot \ln(T)} \cdot \left(1 + \frac{\alpha_a^2}{\epsilon^2}\right) + 8 \cdot N + \sum_{i=1}^{k} p_{i,\epsilon} \cdot T$$

*for $p_{i,\epsilon} = P_{X_0 \sim \mathcal{D}_0}[X_0 \leq \mu_0 - (1 - \epsilon) \cdot \Delta_i]$.*

*Proof.* The first part of this proof is identical to that of Theorem 1. First, we will divide the two arms into groups (group 1 with low-regret arms, group 2 with high regret arms). For arms from group 1, we've shown that we can bound:

$$\sum_{i \in G_1} n_{i,T} \cdot \Delta_i \leq \sqrt{N \cdot T \cdot \ln(T)}$$

For arms from group 2, we work to bound the number of times that arm $i$, for $i \neq 0$, can be pulled for all time ($n_{i,T}$). We will use $n_{i,t}^* = \alpha_a^2 \cdot \frac{\ln(t)}{\epsilon^2 \cdot \Delta_i^2}$ to denote a minimum threshold of samples we need from arm $i$ in order to obtain certain high-probability guarantees.

$$\mathbb{E}[n_{i,T}] = \mathbb{E}\left[\sum_{t=0}^{T} \mathbb{1}(I_{t+1} = i)\right]$$

$$\leq n_{i,T}^* + \mathbb{E}\left[\sum_{t=N}^{T} \mathbb{1}\left(I_{t+1} = i | n_{i,t} \geq n_{i,t}^*\right)\right]$$

$$= n_{i,T}^* + \sum_{t=N}^{T} P\left(I_{t+1} = i | n_{i,t} \geq n_{i,t}^*\right)$$

This proof differs from Theorem 1 because the human selects myopically, greedily optimizing whichever arm has $\hat{\mu}_i$ maximized. Let $X_0 \sim \mathcal{D}_0$ denote the first pull from each arm, respectively. We will use

$$p_{i,\epsilon} = P_{X_0 \sim \mathcal{D}_0}[X_0 \leq \mu_0 - (1 - \epsilon) \cdot \Delta_i]$$

We know with high probability that $\hat{\mu}_{i,t} \leq \mu_i + \epsilon \cdot \Delta_i$, but if $X_0 \leq \mu_0 - (1 - \epsilon) \cdot \Delta_i = \mu_i + \epsilon \cdot \Delta_i$, then it is possible that $X_0 \leq \hat{\mu}_{i,t}$. If this occurs, arm $i$ will always be preferred over arm 0, so arm 0 would never be pulled again, after the first time.

With probability $1 - p_{i,\epsilon}$, $X_0 > \mu_0 - (1 - \epsilon) \cdot \Delta_i = \mu_i + \epsilon \cdot \Delta_i$. If this occurs, with probability at least $1 - \frac{2}{t^2}$, the following inequalities hold:

$$\hat{\mu}_{i,t} \leq \mu_i + \sqrt{\frac{\ln(t)}{n_{i,t}}} \qquad \text{Chernoff bound with probability } 1 - \frac{2}{t^2}$$

$$\leq \mu_i + \epsilon \cdot \Delta_i \qquad |\hat{\mu}_{i,t} - \mu_i| \leq \sqrt{\frac{\ln(t)}{n_{i,t}}} \leq \sqrt{\frac{\ln(t) \cdot \Delta_i^2 \cdot \epsilon^2}{\alpha_a^2 \cdot \ln(t)}} \leq \epsilon \cdot \Delta_i$$

$$= \mu_0 - (1 - \epsilon) \cdot \Delta_i$$
$$\leq X_0 \qquad \text{by assumption}$$

Because $n_{i,T}^* \geq \alpha_a^2 \cdot \frac{\ln(t)}{\Delta_i^2}$, we know that with high probability the algorithm will rank arm 0 above arm $i$. This fails to occur with probability no more than $\frac{2}{t^2}$. Overall, the number of times arm $i$ could be pulled is upper bounded by:

$$\sum_{i=N}^{T} p_{i,\epsilon} + (1 - p_{i,\epsilon}) \cdot \frac{4}{t^2} \leq p_{i,\epsilon} \cdot T + \sum_{t=N}^{T} \frac{4}{t^2} \leq p_{i,\epsilon} \cdot T + 8 \cdot (1 - p_{i,\epsilon})$$

We can bound the instance-independent regret from this term (for arms in group 2):

$$\sum_{i \in G_2} n_{i,t} \cdot \Delta_i \leq \left( \frac{\ln(T)}{\epsilon^2 \cdot \Delta_i^2} + p_{i,\epsilon} \cdot T + 8 \cdot (1 - p_{i,\epsilon}) \right) \cdot \Delta_i$$

$$= \sum_{i \in G_2} \frac{\ln(T)}{\epsilon^2 \cdot \Delta_i} + p_{i,\epsilon} \cdot T \cdot \Delta_i + 8 \cdot (1 - p_{i,\epsilon}) \cdot \Delta_i$$

$$\leq \sum_{i \in G_2} \frac{\alpha_a^2 \cdot \ln(T) \cdot \sqrt{T}}{\epsilon^2 \cdot \sqrt{N \cdot \ln(T)}} + p_{i,\epsilon} \cdot T + 8$$

$$\leq \frac{\alpha_a^2}{\epsilon^2} \cdot \sqrt{\ln(T) \cdot T \cdot N} + 8 \cdot N + \sum_{i \in G_2} p_{i,\epsilon} \cdot T$$

For the last component, we can actually improve the linear dependence slightly. We know that $k$ arms are shown to the human. Therefore, even if $X_0 < \mu_i + \epsilon \Delta_i$ for more than $k$ arms, it won't increase regret by more than the total probability for the $k$ largest probabilities. Because $p_{i,\epsilon}$ is defined relative to distribution $\mathcal{D}_0$, this will be the arms $i$ with smallest $\Delta_i$. The upper bound becomes:

$$\leq \frac{\alpha_a^2}{\epsilon^2} \cdot \sqrt{\ln(T) \cdot T \cdot N} + 8 \cdot N + \sum_{i=1}^{k} p_{i,\epsilon} \cdot T$$

Combined with the regret from the group 1 arms, this gives us regret:

$$\sqrt{N \cdot T \cdot \ln(T)} \cdot \left( 1 + \frac{\alpha_a^2}{\epsilon^2} \right) + 8 \cdot N + \sum_{i=1}^{k} p_{i,\epsilon} \cdot T$$

$\square$

**Lemma 3.** *If $p_{i,\epsilon} > 0$ for any $i$ and $k \geq 2$, regret is $\Omega(T)$.*

*Proof.* Then, we will show it is possible to construct a case with linear regret. Suppose that $\mathcal{D}_j$ for $j \neq 0 \colon X_j \sim \mathcal{D}_j = \mu_j$ with probability 1. Suppose that $k \geq 2$, so the algorithm always must present at least one other arm besides the optimal arm. Consider any arm $i \neq 0$. Then, with probability $p_{i,\epsilon}$, after 1 samples $\hat{\mu}_0 = X_0 < \mu_0 + (1 - \epsilon) \cdot \Delta_i$. If this occurs, then a myopic human will always select arm $i$ rather than arm 0. Because arm $i$ is deterministic, $\hat{\mu}_i$ will not update, but will remain greater than $\mu_0$, which means arm 0 will never be pulled again. This leads to regret $p_{i,\epsilon} \cdot \Delta_i \cdot T$. $\square$

Collectively, these results give us an upper bound on expected regret for the human-algorithm MAB setting. Additionally, they demonstrate areas where we should not linear regret is unavoidable.

## 5 EXPERIMENTAL RESULTS

Next, in this section we further explore this setting through simulations. Specifically, our goal will be to demonstrate how expected regret varies with different features, such as the number of arms that are presented to the human, or the distribution of rewards for each arm. In each simulation, we have $N = 5$ arms and varying $k \in [1, N]$. There is a single best arm 0 with highest expected reward drawn $\mathcal{N}(\mu_0, \sigma)$, while all other arms have identical rewards $\mathcal{N}(\mu_i, \sigma)$ (unless otherwise noted). We use rewards drawn from normal distributions because varying $\sigma$ will allow us to change $p_{i,\epsilon}$ while keeping the relative expected means $\mu_0, \mu_i$ the same. We fix the algorithm's exploration coefficient $\alpha_a = 1$ throughout, but vary the human's exploration coefficient $\alpha_h$.

### 5.1 VARYING NUMBER OF ARMS PRESENTED

In Figure 1, we explore the effect of varying the human's exploration coefficient $\alpha_h$ for values in $\{0, 0.5, 1, 2\}$. Note that for $\alpha_h = 0$ (Figure 1a), linear regret dominates and is increasing in the number of arms presented to the human $k$, as would be expected from Theorem 2. As intuition,

consider the fact that for larger $k$ the human has a higher probability of being presented with an arm $i$ for which the first sample $X_0 < \mu_0 - (1 - \epsilon) \cdot \Delta_i$. Therefore, showing more arms increases the linear component of regret.

For $\alpha_h > 0$ regret is sublinear in $T$. For Figure 1b with $\alpha_a > \alpha_h > 0$, regret is *decreasing* in $k$: showing more arms to the human decreases regret. Obviously, for Figure 1c with $\alpha_h = \alpha_a$ regret is constant in $k$ because the human and algorithm are using the same metric to decide which arm to pick. Finally, for Figure 1d with $\alpha_h > \alpha_a$, regret is again *increasing* in $k$. As intuition, it might be useful to recall that the standard UCB algorithm Auer et al. (2002) has regret that is increasing in $\alpha$, for $\alpha > 0$. If $\alpha_h < \alpha_a$, then for larger $k$ the human is doing "more" of the selection, because it has more arms to choose from. Because of this, the overall regret will be dominated by the human's, with a coefficient of $\alpha_h$. Conversely, if $\alpha_h > \alpha_a$, then for larger $k$ regret will be higher, again because the human is "responsible" for choosing among a larger set of items.

These experimental results indicate that it could be possible to find a version of Theorem 1 that also depends on $k$. However, this attempt might be complicated by the non-linear relationship between regret and $k$. For Figure 1b, for example, there is a large jump in regret from $k = 5$ to $k = 4$, but a much smaller one between $k = 4$ and $k = 3$. Interestingly, these experimental results also seem to indicate that *complementarity* (strict improvements through human-algorithm collaboration) might be impossible. For every simulation shown, either the human alone ($k = N$) or the algorithm alone ($k = 1$) performs optimally.

## 5.2 VARYING REWARD DISTRIBUTION

Finally, we explored the impact of varying the reward distribution. Figure 2 shows another example with $\alpha_h = 0$, but where regret is sublinear, as compared with Figure 1a, which has $\alpha_h = 0$, but linear regret. In Figure 2, the gap $\Delta_i$ is larger, so $p_{i,\epsilon}$ is extremely small. This means that the linear dependence on $T$ is very small, so regret is dominated by the sublinear term. This example shows (empirically) that it is possible to achieve low regret, given certain distributional assumptions on the arms.

## 6 CONCLUSION AND FUTURE DIRECTIONS

In this paper, we have explored human-algorithm collaboration in a multi-armed bandit scenario. We proved theoretical bounds on regret, as well as demonstrated empirical patterns in regret when varying number of arms are presented. There are multiple possible avenues for future work. For example, in our work we have assumed that the algorithm is using UCB. While this is a standard MAB approach that achieves sublinear regret with a human collaborator in many cases, it may not be optimal. It would be interesting to explore whether another algorithm could achieve superior performance for a range of models of human behavior. In particular, so far results seem to imply that complementarity is impossible with the current setting. It would be useful to know whether this limitation is inherent to the setting or if regret could be improved to achieve complementarity. Relatedly, Section 5 shows intriguing experimental patterns in average regret, $k$, and $\alpha_a, \alpha_h$ that could be useful to explore theoretically. Finally, it would be interesting to explore cases where the human and algorithm have access to different historical sets of rewards data (reflecting cases where the human, for example, may have lived in the city for a while, and so has access to prior information about the best routes).

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

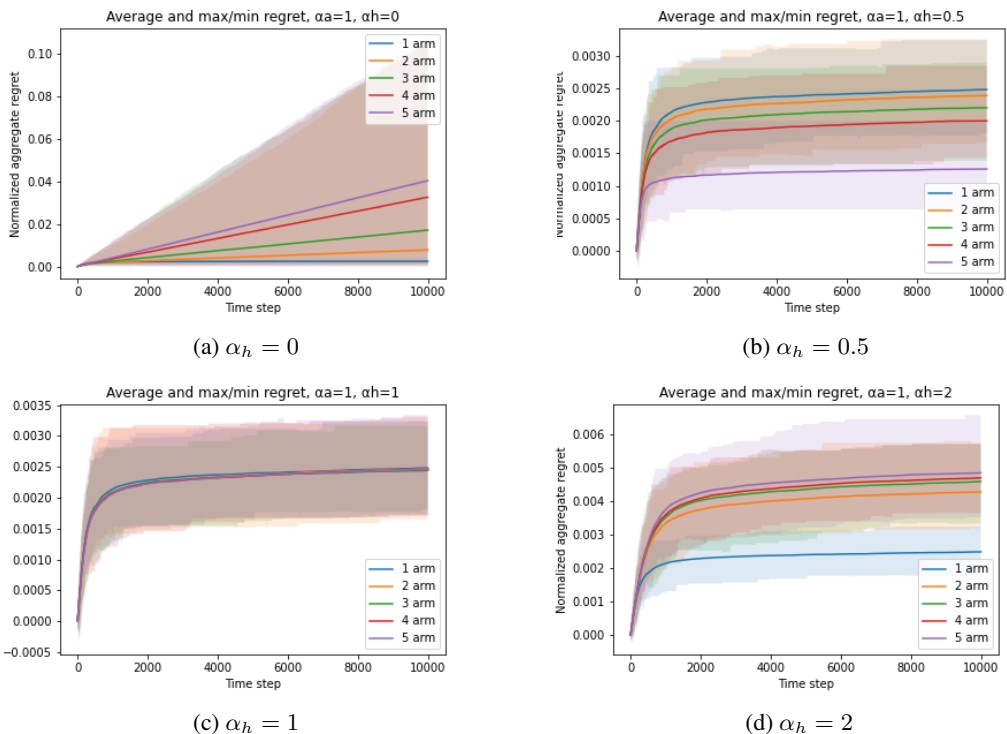

(a) $\alpha_h = 0$

(b) $\alpha_h = 0.5$

(c) $\alpha_h = 1$

(d) $\alpha_h = 2$

Figure 1: Plots show regret for (line shows average regret, shaded region is max and min regret, for 100 simulations, each with 10,000 time steps). Each simulation has 5 arms total, where the single best arm has reward drawn from $\mathcal{N}(\mu = 0.5, \sigma = 0.1)$ and the other arms have reward $\mathcal{N}(\mu = 0.45, \sigma = 0.1)$.

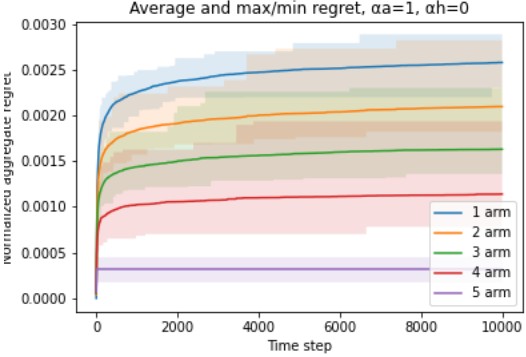

Figure 2: Simulation identical to Figure 1a 1, except that, while the best arm still has reward drawn from $\mathcal{N}(\mu = 0.5, \sigma = 0.1)$, the other arms now have reward drawn from $\mathcal{N}(\mu = 0.1, \sigma = 0.1)$. The larger gap in rewards means that $p_{i,\epsilon}$ is smaller, so the linear dependence on $T$ does not dominate overall regret.

Gagan Bansal, Tongshuang Wu, Joyce Zhou, Raymond Fok, Besmira Nushi, Ece Kamar, Marco Tulio Ribeiro, and Daniel Weld. Does the whole exceed its parts? The effect of AI explanations on complementary team performance. In *Proceedings of the 2021 CHI Conference on Human Factors in Computing Systems*, pp. 1–16, 2021b.

Hamsa Bastani, Pavithra Harsha, Georgia Perakis, and Divya Singhvi. Learning personalized product recommendations with customer disengagement. *Manufacturing & Service Operations Management*, 24(4):2010–2028, 2022.

Emma Beede, Elizabeth Baylor, Fred Hersch, Anna Iurchenko, Lauren Wilcox, Paisan Ruamviboonsuk, and Laura M Vardoulakis. A human-centered evaluation of a deep learning system deployed in clinics for the detection of diabetic retinopathy. In *Proceedings of the 2020 CHI Conference on Human Factors in Computing Systems*, pp. 1–12, 2020.

Sebastian Bordt and Ulrike Von Luxburg. A bandit model for human-machine decision making with private information and opacity. In *International Conference on Artificial Intelligence and Statistics*, pp. 7300–7319. PMLR, 2022.

Lawrence Chan, Dylan Hadfield-Menell, Siddhartha Srinivasa, and Anca Dragan. The assistive multi-armed bandit. In *2019 14th ACM/IEEE International Conference on Human-Robot Interaction (HRI)*, pp. 354–363. IEEE, 2019.

Alexandra Chouldechova, Diana Benavides-Prado, Oleksandr Fialko, and Rhema Vaithianathan. A case study of algorithm-assisted decision making in child maltreatment hotline screening decisions. In *Conference on Fairness, Accountability and Transparency*, pp. 134–148. PMLR, 2018.

Bo Cowgill and Megan T Stevenson. Algorithmic social engineering. In *AEA Papers and Proceedings*, volume 110, pp. 96–100, 2020.

Ruijiang Gao, Maytal Saar-Tsechansky, Maria De-Arteaga, Ligong Han, Min Kyung Lee, and Matthew Lease. Human-ai collaboration with bandit feedback. *arXiv preprint arXiv:2105.10614*, 2021.

Xinyan Hu, Dung Daniel Ngo, Aleksandrs Slivkins, and Zhiwei Steven Wu. Incentivizing combinatorial bandit exploration. *arXiv preprint arXiv:2206.00494*, 2022.

Nicole Immorlica, Jieming Mao, Aleksandrs Slivkins, and Zhiwei Steven Wu. Incentivizing exploration with selective data disclosure. *arXiv preprint arXiv:1811.06026*, 2018.

Sampath Kannan, Michael Kearns, Jamie Morgenstern, Mallesh Pai, Aaron Roth, Rakesh Vohra, and Zhiwei Steven Wu. Fairness incentives for myopic agents. In *Proceedings of the 2017 ACM Conference on Economics and Computation*, pp. 369–386, 2017.

Junpei Komiyama, Junya Honda, Hisashi Kashima, and Hiroshi Nakagawa. Regret lower bound and optimal algorithm in dueling bandit problem. In *Conference on learning theory*, pp. 1141–1154. PMLR, 2015.

Tor Lattimore and Csaba Szepesvári. *Bandit algorithms*. Cambridge University Press, 2020.

Sarah Lebovitz, Hila Lifshitz-Assaf, and Natalia Levina. To incorporate or not to incorporate ai for critical judgments: The importance of ambiguity in professionals' judgment process. *Collective Intelligence, The Association for Computing Machinery*, 2020.

Sarah Lebovitz, Natalia Levina, and Hila Lifshitz-Assaf. Is AI ground truth really "true"? the dangers of training and evaluating AI tools based on experts' know-what. *Management Information Systems Quarterly*, 2021.

David Madras, Toni Pitassi, and Richard Zemel. Predict responsibly: Improving fairness and accuracy by learning to defer. In S. Bengio, H. Wallach, H. Larochelle, K. Grauman, N. Cesa-Bianchi, and R. Garnett (eds.), *Advances in Neural Information Processing Systems*, volume 31. Curran Associates, Inc., 2018. URL https://proceedings.neurips.cc/paper/2018/file/09d37c08f7b129e96277388757530c72-Paper.pdf.

Chinasa T Okolo, Srujana Kamath, Nicola Dell, and Aditya Vashistha. "it cannot do all of my work": Community health worker perceptions of ai-enabled mobile health applications in rural india. In *Proceedings of the 2021 CHI Conference on Human Factors in Computing Systems*, pp. 1–20, 2021.

Maithra Raghu, Katy Blumer, Greg Corrado, Jon Kleinberg, Ziad Obermeyer, and Sendhil Mullainathan. The algorithmic automation problem: Prediction, triage, and human effort. *NeurIPS Workshop on Machine Learning for Health (ML4H)*, 2018.

Charvi Rastogi, Liu Leqi, Kenneth Holstein, and Hoda Heidari. A unifying framework for combining complementary strengths of humans and ml toward better predictive decision-making. *arXiv preprint arXiv:2204.10806*, 2022.

Aadirupa Saha and Aditya Gopalan. Battle of bandits. In *UAI*, pp. 805–814, 2018.

Pannaga Shivaswamy and Thorsten Joachims. Coactive learning. *Journal of Artificial Intelligence Research*, 53:1–40, 2015.

Aleksandrs Slivkins et al. Introduction to multi-armed bandits. *Foundations and Trends® in Machine Learning*, 12(1-2):1–286, 2019.

Mark Steyvers, Heliodoro Tejeda, Gavin Kerrigan, and Padhraic Smyth. Bayesian modeling of human–ai complementarity. *Proceedings of the National Academy of Sciences*, 119(11): e2111547119, 2022.

Eleni Straitouri, Lequn Wang, Nastaran Okati, and Manuel Gomez Rodriguez. Provably improving expert predictions with conformal prediction, 2022.

Yanan Sui, Yisong Yue, and Joel W Burdick. Correlational dueling bandits with application to clinical treatment in large decision spaces. *arXiv preprint arXiv:1707.02375*, 2017a.

Yanan Sui, Vincent Zhuang, Joel W Burdick, and Yisong Yue. Multi-dueling bandits with dependent arms. *arXiv preprint arXiv:1705.00253*, 2017b.

Yanan Sui, Masrour Zoghi, Katja Hofmann, and Yisong Yue. Advancements in dueling bandits. In *IJCAI*, pp. 5502–5510, 2018.

Qian Yang, Alex Scuito, John Zimmerman, Jodi Forlizzi, and Aaron Steinfeld. Investigating how experienced ux designers effectively work with machine learning. In *Proceedings of the 2018 Designing Interactive Systems Conference*, pp. 585–596, 2018.

Yisong Yue, Josef Broder, Robert Kleinberg, and Thorsten Joachims. The k-armed dueling bandits problem. *Journal of Computer and System Sciences*, 78(5):1538–1556, 2012.

## A  SUPPLEMENTARY PROOFS

**Lemma 1.** *Any human that selects uniformly at randomly with probability $\delta > 0$ among $k \geq 2$ arms incurrs regret $\Omega(T)$ that is increasing in $k$.*

*Proof.* Suppose that $k$ arms are presented and the human selects randomly with probability $\delta$. Each of the presented arms has at least a $\frac{1}{k} \cdot \delta$ probability of being selected. This chance could be higher, because with probability $1 - \delta$ the human selects according to some other strategy, but $\frac{1}{k} \cdot \delta$ is a lower bound. We can similarly lower bound regret by assuming that the algorithm (in each round) is optimal and selects the top $k$ arms to present to the human.

Then, expected regret in a given round is lower bounded by by:

$$\delta \cdot \frac{1}{k} \sum_{i=2}^{k} \Delta_i \geq \delta \cdot \frac{k-1}{k} \cdot \Delta_2$$

which gives an overall lower bound on regret by given by:

$$T \cdot \delta \cdot \frac{k-1}{k} \cdot \Delta_2$$

For $\delta > 0, k \geq 2$, this gives regret that is linear in $T$. □

**Lemma 2.** *Consider arm $i$ with $n_{i,t} \geq \alpha^2 \cdot \frac{\ln(t)}{\epsilon^2 \cdot \Delta_i^2}$, for $\epsilon \in [0, 0.5], \alpha \geq 1$. Then, if any UCB algorithm selecting according to $UCB_{i,t} = \hat{\mu}_{i,t} + \alpha \cdot \sqrt{\frac{\ln(t)}{n_{i,t}}}$. Then, $UCB_{i,t} < UCB_{0,t}$ with probability at least $1 - \frac{4}{t^2}$.*

*Proof.*

$$UCB_{i,t} = \hat{\mu}_{i,t} + \alpha \cdot \sqrt{\frac{\ln(t)}{n_{i,t}}}$$

$$\leq \hat{\mu}_{i,t} + \alpha \cdot \sqrt{\frac{\ln(t) \cdot \epsilon^2 \cdot \Delta_i^2}{\alpha^2 \cdot \ln(t)}} \quad n_{i,t} \geq \alpha^2 \cdot \frac{\ln(t)}{\epsilon^2 \cdot \Delta_i^2}$$

$$< \hat{\mu}_{i,t} + \epsilon \cdot \Delta_i$$

$$< \mu_i + \epsilon \cdot \Delta_i + \epsilon \cdot \Delta_i \qquad |\hat{\mu}_{i,t} - \mu_i| \leq \sqrt{\frac{\ln(t)}{n_{i,t}}} \leq \sqrt{\frac{\ln(t) \cdot \Delta_i^2 \cdot \epsilon^2}{\alpha^2 \cdot \ln(t)}} \leq \epsilon \cdot \Delta_i \text{ with probability } 1 - \frac{2}{t^2}$$

$$= \mu_0 \qquad \epsilon \leq 0.5$$

$$< UCB_{0,t} \qquad \mu_i \leq \hat{\mu}_{i,t} + \alpha \cdot \sqrt{\frac{\ln(t)}{n_{i,t}}} \text{ with probability } 1 - \frac{2}{t^2}$$

Overall, this inequality holds with probability at least $1 - \frac{4}{t^2}$ □

