# OpenReview forum: "'I pick you choose': Joint human-algorithm decision making in multi-armed bandits"
_ICLR.cc/2023/Conference — Submitted to ICLR 2023_

### Official Review · Reviewer_c8Yy · 2022-10-17

**Confidence:** 4
**Correctness:** 3
**Technical Novelty And Significance:** 2
**Empirical Novelty And Significance:** Not applicable
**Recommendation:** 3

**Clarity, Quality, Novelty And Reproducibility:**

Some concerns about the proof of Theorem 2.

What do you mean by the last inequality in Page 6? I can understand that when the first pull of arm 0 is less than that threshold, then the regret can be linear with $T$. But what if the first pull of arm 0 is larger than that threshold, while the second pull of arm 0 is bad enough so that the empirical mean is then smaller than that threshold?

I think you may need to define $p_{i,\epsilon}$ as the probability of "the empirical mean of arm 0 is less than that threshold in some time step".

**Strength And Weaknesses:**

Strength

1. The human-algorithm collaborative framework plays an important role in machine learning researches.

Weakness

1. The algorithm seems to be useless in the system.

When $\alpha_h = 0$, it is showed that the regret is linear with $T$, even if there is an algorithm that controls the candidate arms. Though this regret can be better than the only-human case, we do not want to regard it as an efficient framework.

When $\alpha_h \ge 1$, according to Theorem 1, the algorithm can only make the regret upper bound worse (or at least the improvement is limited).

Then my question is, why we must use such a human-algorithm collaborative framework is these examples? Why not get rid of the algorithm and only let the human make decisions?

2. The theoretical contribution of this paper is limited.

The theorems and lemmas in this paper seem trivial. I read the full proofs, but I do not see any new techniques or novel insights within them. They are just tranditional UCB-based analysis and there seem to be no technical challenges.



**Summary Of The Paper:**

This paper considers a human-algorithm collaborative framework in bandit problems. In this framework, the algorithm will first pick several candidate arms based on its picking rule, and then the human will choose one arm in this candidates to pull, based on his/her choosing rule. In this paper, the authors mainly consider the case that both the human and the algorithm are running UCB-type algorithms, i.e., their decisions are made based on the upper confidence bounds of all the arms (with different parameter $\alpha_h$ and $\alpha_a$). In this case, they show that when both $\alpha_a$ and $\alpha_h$ are large enough (e.g., larger than 1), then the regret of the system is at most the same as the worse one (either the human or the algorithm). On the other hand, if $\alpha_h = 0$ (i.e., the human is totally greedy), then the regret of this system is linear with $T$, as long as the algorithm must provide at least 2 candidate arms to the human. Finally, the authors use some experimental results to demonstrate their theoretical findings.

**Summary Of The Review:**

Overall, I do not think the contribution of this paper is enough for an ICLR submission.

---

> ### Author Response · Authors · 2022-11-17
> **Response to c8Yy**
>
> Some of the comments you raised were addressed in “Comments to all reviewers”, above. One question you asked was why the algorithm would be included, rather than just the human alone, given Theorem 1’s regret bounds. There are two possible aspects to this answer: the first is that, though Theorem 1 gives an upper bound on regret, we see empirically in Section 5 that for certain settings we do see improvements in the joint human-algorithm system. The second answer is more general: it might be, for example, that the algorithm is necessary for a reason orthogonal to learning. For example, the routing example at the start of the paper suggested a case where the algorithm is suggesting driving routes to a human. Here, the algorithm may be necessary because the human may simply not know the routes, or because N (total number of routes) is infeasible for them to sort through.
>
>
> Finally, you had a specific question about $p_{i, \epsilon}$ and whether it might be better defined as $p_{i, \epsilon}(s)$, being the probability that the average of the first $s$ samples from that arm is less than $\mu_0 - (1-\epsilon) \cdot \Delta_i$. In order to consider this point, it is useful to consider the purpose of $p_{i, \epsilon}$. Its role is to upper bound the probability that the empirical mean of the best arm $\hat \mu_{0, t}$ falls below $1-\epsilon \cdot \Delta_i$ - that is, “too close” to the true mean of the $i$th arm. As you noted, if the best arm 0 has been pulled multiple times, then the empirical mean $\hat \mu_{0, t}$ will be made up of multiple samples, so in general $p_{i, \epsilon}(s)<=p_{i, \epsilon}(0)$ (because an average of multiple iid samples is more likely to be close to its mean than a single sample). Thus, the use of $p_{i, \epsilon}(0)$ is an upper bound and is likely not tight. (Note that while we use $1-p_{i, \epsilon}$, we upper bound this with 1, which preserves the overall upper bound). However, in order to make it tight, we would need to reason about the number of times arm 0 has been pulled at the time that arm $i$ is presented, which might be intractable. Even if this bound were tightened, it would still retain the linear dependence on $T$, but simply with a coefficient involving $p_{i, \epsilon}(s)>0$. Thank you for this detailed question - this is a somewhat subtle point, so we will include a clear explanation in a revised version of this paper.

---

> > ### Comment · Reviewer_c8Yy · 2022-11-21
> > **Reply to the rebuttal**
> >
> > The authors mentioned that one reason for applying the algorithm-human framework is that there are some empirically improvements.
> > I think there should be more analysis for this point. For example, in the case that $\alpha_h > \alpha_a$ and the algorithm only provide 2 candidates to the human, then for the arms $i\ge 2$, it is posible that after ${\alpha_a\log T \over \Delta_i^2}$ number of pulls, the algorithm will not provide them to the human as candidates, therefore, the regret upper bound now could becomes ${\alpha_h\log T \over \Delta_1} + \sum_{i\ge 2}{\alpha_a\log T \over \Delta_i} $, which is better than human running his own policy ($\sum_{i\ge 1}{\alpha_h\log T \over \Delta_i} $).
> >
> >
> >
> > For the definition about $p_{i.\epsilon}$. Assume that the rewards are Bernoulli, $\mu_0 = 0.99$ and $\mu_0 - (1-\epsilon)\Delta_i = 0.9$. Then we know that $p_{i,\epsilon}(1) = 0.01$, and $p_{i,\epsilon}(2) = 0.0199$, and in this way $p_{i,\epsilon}(2) > p_{i,\epsilon}(1)$. Besides, even if $p_{i,\epsilon}(2) < p_{i,\epsilon}(1)$, it is not correct to say that using $p_{i,\epsilon}(1)$ is enough (at least this is not enough in your analysis). You may "stuck" at any time point, therefore you need a bound for the probability of "stucking" anywhere but not only in some fixed steps.

---

> > > ### Author Response · Authors · 2022-11-21
> > > **Comments on response to rebuttal**
> > >
> > > Thank you for your follow-up notes!
> > >
> > >  Your first note described the potential for a tighter theoretical bound, potentially by saying that after ${\alpha_a\log T \over \Delta_i^2}$ samples have been observed, the algorithm might be unlikely to present sub-optimal arms (the bottom $N-k$ arms) to the human. This is an intruiging idea, and one we had explored in previous versions of this proof. Unfortunately, the idea as stated doesn't necessarily hold. For example, suppose that all arms $i \ne 0$ have equal true means (as in Figure 1, our empirical exploration). If the algorithm has ${\alpha_a\log T \over \Delta_i^2}$ samples for each of them, then with high probability $\hat UCB_0^a > \hat UCB_i^a$. However, we cannot make a similar high-probability statement for whether $\hat UCB_i^a > \hat UCB_j^a$ for $i, j \ne 0$, which is what we would need in order to say that the algorithm won't select certain arms to be presented to the human. However, this type of argument would likely work if we assume a nonzero gap between each arms, e.g. $\Delta_i>  \Delta_j $ $\forall i<j$. This assumption is not necessary for empirical improvement, since Figure 1 has $\Delta_i = \Delta_j$ for $i, j \ne 0$. However, the empirical benefits in Figure 1 can be quite small: note that in Figure 1b, 1d the highest curve (regret of worst of human or algorithm, the theoretical upper bound on regret) is quite close to the second highest curve (regret of combined system, when $k=2$ arms are shown). While it might be possible to get a slightly lower theoretical bound in this setting, it would likely be more straightforward to get a tighter bound if we assume $\Delta_i> \Delta_j$, which is an avenue we're actively exploring.
> > >
> > > Your second point had to do with $p_{i, \epsilon}(s)$. Thank you for your clear and insightful counter-example showing that this term might be increasing in $s$! We will update the proof to reflect this fact. For simplicity, we may simply define $p_{i, \epsilon} = \max_{s\in [T]} p_{i, \epsilon}(s)$, which would increase the constant multiplicative factor on $T$ but not qualitatively change our results.

---

> > > > ### Comment · Reviewer_c8Yy · 2022-11-22
> > > > **Reply**
> > > >
> > > > It is also worth to state the result (e.g., assuming a non-zero gap between every arms) that applying the algorithm can make the human achieve better perfermance theoretically. This could be one of the technically new points in your paper.
> > > >
> > > > On the other hand, I am wondering whether the empirically improvement in the case that $\Delta_i = \Delta_j$ (i.e., Figure 1. (d)) is upper bounded by some constant. It seems that the lines $n=2,3,4,5$ are very close. What would the regret lines look like if the $\Delta_i$'s are not the same?

---

> > > > > ### Author Response · Authors · 2022-11-29
> > > > > **Reply**
> > > > >
> > > > > Thanks so much for your question! We agreed that it was interesting, so we ran some more simulations. These can be seen at this anonymous link (https://anonymous.4open.science/r/iclr_rebuttal-07BC/). We ran simulations first varying the gap $\Delta_i$ (with $\mu_i \in \{0.5, 0.4, 0.3, 0.2, 0.1\}$) (Figure 3), then by shrinking the variance $\sigma$ from 0.1 to 0.01 (Figure 4), and then changing the gap $\Delta_i$ to be $\mu_i \in \{0.5, 0.45, 0.4, 0.35, 0.3\}$ (Figure 5). Interestingly, the average regret in each case seemed to stay roughly the same. However, the maximum and minimum regret (the shaded regions) grew narrower (which makes sense given smaller $\sigma$). This area seems to warrant future investigation, which we will work on exploring.

---

### Official Review · Reviewer_DhuT · 2022-10-24

**Confidence:** 4
**Correctness:** 4
**Technical Novelty And Significance:** 2
**Empirical Novelty And Significance:** 1
**Recommendation:** 1

**Clarity, Quality, Novelty And Reproducibility:**

The paper is clear, and the novelty in terms of the setting is evident.
The authors did not provide the code for the experiments, even if they are described in detail.

**Strength And Weaknesses:**

I think that the motivating example provides an interesting setting in which the research is still somehow missing a deeper analysis. However, the arguments and analysis of the authors do not seem to be complete enough yet for publication at ICLR. In my opinion, the author should focus more on the setting, for instance analysing the lower bound on the regret before proposing algorithms.

Moreover, I think that the regret defined by the authors does not reflect the fact that the user might have different goals. I think that the modeling in this aspect may be revised and improved.

Finally, the experimental results are not convincing at all. Indeed, they are on a too narrow set of settings, and statistical significance is not provided in most of the cases.

Lemma 1: Increasing in K is a bit generic. What is the dependence on this parameter?



Minor:
Remove all the contractions like isn't and doesn't

I think that in the literature review, you should also mention and discuss the corrupted bandit setting. For instance,

Gajane, Pratik, Tanguy Urvoy, and Emilie Kaufmann. "Corrupt bandits." EWRL (2016).

Zhao, Heyang, Dongruo Zhou, and Quanquan Gu. "Linear contextual bandits with adversarial corruptions." arXiv preprint arXiv:2110.12615 (2021).

and literature therein cited.

Define a symbol for regret.

Please provide the comments on Theorem 1 after it has been stated.

Theorem 1: It is not clear to me where did you use the fact that \alpha_h > 1 and \alpha_a > 1

Theorem 2: Does this regret match the theoretical lower bound? Is it possible to show that this is the best one might do in this setting?

Assuming the shaded areas are the confidence intervals for the analysed mean regret (it is not clearly stated in the text), I think that the results provided are not statistically significant.

**Summary Of The Paper:**

The paper is about a model of sequential decision-making in which the algorithm is only allowed to select a subset of the possible options, and a human user is then required to choose one among them. The authors analysed theoretically the case in which both the agents (algorithm and human) are both relying on a Hoeffding-like bound to select the options. Finally, they present some synthetically generated experiments to evaluate the performance of the UCB-like algorithm when different numbers of arms are shown to the user.

**Summary Of The Review:**

The paper presents a novel setting, but the authors did not succeed in providing a deep analysis of it.

---

> ### Author Response · Authors · 2022-11-17
> **Response to DhuT**
>
> Some of the comments you raised were addressed in “Comments to all reviewers”, above. More specifically, one question you raise is about lower bounds on regret. One result our paper has in this vein is Lemma 1, which shows a linear dependence on T is unavoidable when the human picks randomly with any positive probability. It would be interesting to explore lower bounds on regret for other cases, such as analogs of Theorems 1 and 2. Given our experimental results in Section 5, it wouldn’t be possible to find matching lower bounds, since some examples in Figure 1 show cases where the combined human-algorithm system has lower regret than the upper bound would imply.
>
> More specific comments:
> - For Lemma 1, the dependence is $\frac{k-1}{k}$ (also presented in the proof in the Apppendix).
> - You asked about where the requirement that $\alpha_h >1, \alpha_a >1$ is used. It is used through Theorem 2’s reliance on Lemma 2 (high probability bounds), where it appears in line 4 of the proof.
> - You asked about the shaded regions in the plots. As described in the caption, the shaded region reflects the maximum and minimum regret for each experiment. Because we ran 100 simulations for each of the $k$ values, the maximum and minimum regret range is quite large. We chose to present this range, rather than standard deviation, because it matched more closely with our goals of bounding regret. However, it would be straightforward for a future version of this paper to also include a reproduction of these plots with standard deviation values shown.
> - Thank you for the references!

---

### Official Review · Reviewer_PnxP · 2022-10-24

**Confidence:** 4
**Clarity, Quality, Novelty And Reproducibility:** Both the writing and the content of t…
**Correctness:** 1
**Technical Novelty And Significance:** 1
**Empirical Novelty And Significance:** Not applicable
**Recommendation:** 1

**Strength And Weaknesses:**

Strength:
* The problem of human-algorithm coorperation is interesting. The authors provide sufficient literature on such problem.

Weaknesses:
* The problem formulation is quite standard, which provides narrow space to make contributions.
* Section~3.2 only introduces the part of a policy for "human", and I don't see the other part for "algorithm" (or assumptions on how the "algorithm" will behave). So I don't understand what the theoretical analysis and the regret results are for.

**Summary Of The Paper:**

This paper considers the problem of reward maximization in human-algorithm coorperation situation. The paper provides a policy and theoretical analysis. Then the paper shows some experiment results.

**Summary Of The Review:**

This paper is not ready to be published.

---

> ### Author Response · Authors · 2022-11-17
> **Response to PnxP**
>
> Some of the comments you raised were addressed in “Comments to all reviewers”, above. As one note, you mentioned that the formulation seemed too standard. We agree that the formulation seems like a very natural model, which is why we were surprised that an extensive literature review of the bandits literature failed to turn up an analysis of this framework. For example, dueling bandits frameworks assume a fixed human preference model, rather than a learned model. We think that this framework admits natural extensions, but is already natural enough that it justifies analysis.
> Additionally, one correction: you said that only the human’s behavior was described, rather than the algorithm’s. To clarify, the algorithm’s behavior is defined in Section 3.2 (“Because the UCB algorithm is a standard algorithm for multi-armed bandit settings, we will assume the algorithm A uses some variant of it”) and again (more specifically) within each proof (e.g. in Theorem 1, “Recall that the algorithm uses  $UCB_{i, t}^a = \hat \mu_i + \alpha_a \cdot \sqrt{\frac{\ln(t)}{n_{i, t}}}$) . We will make the description in Section 3.2 more explicit so as to be more clear for future readers.

---

### Official Review · Reviewer_Skb5 · 2022-10-25

**Confidence:** 4
**Correctness:** 4
**Technical Novelty And Significance:** 1
**Empirical Novelty And Significance:** 2
**Recommendation:** 3

**Clarity, Quality, Novelty And Reproducibility:**

The considered joint human-algorithm system in multi-armed bandits is a very interesting and novel problem. But the theoretical analysis and techniques in this paper are similar to those in classic bandit works. The authors present experimental results, and do not provide the code.

**Strength And Weaknesses:**

Strengths:

1. The considered joint human-algorithm system in multi-armed bandits is very interesting and well-motivated.

Weaknesses:

1. While the considered joint human-algorithm system in multi-armed bandits is a very interesting and well-motivated problem, the formulation for this human-algorithm system in this paper is too simple. Specifically, the formulation just considers a simple combination of selecting the empirical best arm and selecting the optimistic best arm (according to the classic UCB algorithm). Under this simple formulation, the joint human-algorithm system does not bring too many challenges to the original multi-armed bandit problem.

2. The theoretical analysis and results in this paper are similar to those in the classic multi-armed bandit works, e.g., the upper confidence bound-based analysis and the empirical myopic analysis. It is unclear to me what unique challenges the joint human-algorithm system impose on the theoretical analysis in this paper.

3. The writing of this paper seems causal. There are several blanks and typos.


**Summary Of The Paper:**

This paper considers joint human-algorithm system in multi-armed bandits. The authors explore multiple possible frameworks for human objectives and provide theoretical regret bounds. They also give experimental results which show how regret varies with the human decision-maker’s objective and the number of arms.

**Summary Of The Review:**

This paper considers a very interesting and well-motivated problem, i.e., a joint human-algorithm system in multi-armed bandits. But the formulation considered in this paper is too simple, which makes the theoretical analysis in this paper too standard (very similar to classic UCB analysis). The unique challenge of joint human-algorithm system is not well captured and analyzed in this paper. The writing of this paper looks causal, and needs to be improved. For these reasons, I give rejection.

=========

Thank the authors for their response. I think this problem is well-motivated and worth a deeper analysis. The current formulation and analysis is too simple to fully capture the interesting insights and unique challenges of human-algorithm decision making. I think this work can be significantly improved by using theoretically deeper formulation and analysis. Therefore, I keep my score.

---

> ### Author Response · Authors · 2022-11-17
> **Response to Skb5**
>
> Some of the comments you raised were addressed in “Comments to all reviewers”, above. We agree that it would be interesting to incorporate more aspects of how humans behave into our model, which would be a potential avenue for future work. Additionally, you noted some typos and informal language, which we will revise in future versions.

---

> > ### Comment · Reviewer_Skb5 · 2022-11-22
> > **Reply to the Rebuttal**
> >
> > Thank you for your reply. I think this problem is well-motivated and worth a deeper analysis. The current formulation and analysis is too simple to fully capture the interesting insights and unique challenges of human-algorithm decision making. I think this work can be significantly improved by using theoretically deeper formulation and analysis. I keep my score.

---

### Author Response · Authors · 2022-11-17
**General comments**

First, we wish to thank all of the reviewers for their detailed and thoughtful comments. In this note we will give general responses that multiple reviewers raised, and on each reviewer’s review we will give more specific comments.

Multiple reviewers said that our model was too simple or didn’t capture certain aspects of human-algorithm collaboration, such as differential levels of knowledge. We absolutely agree that human-algorithm systems are complex, involving factors such as diverse human preferences, knowledge, attention spans, participation rates, and more. Modeling all of these factors are intriguing areas for future work, and many of them would likely fit into extensions of our current model. In this current paper, we focus on one aspect of human-algorithm collaboration: different sensitivities to exploration vs exploitation. In our framework, we model the human and the algorithm as differing solely in their exploration parameter (typically denoted as $\alpha$ in UCB algorithms). Despite how natural this model seemed, we were surprised that an extensive literature review failed to produce any prior analysis of this framework, a gap that our paper closes. We were also surprised that the analysis was non-trivial, especially by the non-obvious dependence on the number of items $k$. We agree that further expanding this model, especially to incorporate differential preferences or levels of knowledge for the human, could be an interesting avenue for future work.

---

### Decision · Program_Chairs · 2023-01-20

**Decision:**

Reject

**Justification For Why Not Higher Score:**

All reviewers agree that the actual model in this paper is trivial and uninteresting, though the overall direction could be interesting.

**Justification For Why Not Lower Score:**

N/A

**Metareview: Summary, Strengths And Weaknesses:**

This paper introduces a hybrid human-ML algorithm for the classic multi-armed bandit problem.

All reviewers felt that this general idea was novel and interesting, however the specific model was much too simple to be of theoretical or practical interest.